# The Prevalence of Hypogonadism and the Effectiveness of Androgen Administration on Body Composition in HIV-Infected Men: A Meta-Analysis

**DOI:** 10.3390/cells10082067

**Published:** 2021-08-12

**Authors:** Daniele Santi, Giorgia Spaggiari, Walter Vena, Alessandro Pizzocaro, Mario Maggi, Vincenzo Rochira, Giovanni Corona

**Affiliations:** 1Department of Biomedical, Metabolic and Neural Sciences, University of Modena and Reggio Emilia, 42121 Modena, Italy; daniele.santi@unimore.it; 2Unit of Endocrinology, Department of Medical Specialties, Azienda Ospedaliero-Universitaria of Modena, Ospedale di Baggiovara, 41125 Modena, Italy; giorgia.spaggiari87@gmail.com; 3Endocrinology, Diabetology and Andrology Unit, Humanitas Clinical and Research Center, IRCCS, 20089 Rozzano, Italy; walter.vena@gmail.com (W.V.); pizzocaroalessandro@gmail.com (A.P.); 4Endocrinology Unit, Department of Experimental, Clinical and Biomedical Sciences, University of Florence, 50139 Florence, Italy; m.maggi@dfc.unifi.it; 5Endocrinology Unit, Maggiore-Bellaria Hospital, Medical Department, Azienda USL di Bologna, 40121-40141 Bologna, Italy; jocorona@libero.it

**Keywords:** testosterone, HIV, body composition, androgen replacement treatment, hypogonadism

## Abstract

Background: Hypogonadism is a common comorbidity in human immunodeficiency virus (HIV)-infected men, although the real prevalence is difficult to be estimated. Moreover, in HIV settings, the efficacy of exogenous testosterone (Te) administration at improving body composition remains unclear. Aim of the study: This review has a double aim. First, to estimate the prevalence of pituitary–testis axis abnormality in HIV-infected patients compared to uninfected subjects. Second, to evaluate the effect of androgen administration on body composition in HIV-infected men. Materials and Methods: A systematic review of the literature and meta-analysis was carried out. Two separated literature searches were performed, the first to evaluate the prevalence of Te deficiency in HIV-infected men and the second one to evaluate effects of androgen administration on body composition. Results: The overall prevalence of Te deficiency in HIV-infected men was calculated from 41 studies, showing a 26% prevalence, which was even higher when free T (fT) levels, more than total T, were considered. Indeed, TT serum levels were similar between HIV patients and controls, although higher SHBG and lower fT were detected in HIV populations. When HIV-infected men were treated with exogenous Te, a significant increase in body weight, lean body mass and fat free mass was detected. Conclusion: The systematic review confirms the high prevalence of Te deficiency in HIV-infected men, particularly when fT has been considered. Moreover, chronic androgen supplementation improves body composition, affecting the lean mass compartment. However, considering the general frailty of HIV patients, a tailored indication for Te therapy should be advocated.

## 1. Introduction

Since the early 1990s, the advent of highly active antiretroviral therapy (HAART) drastically changed the natural history of human immunodeficiency virus (HIV) infection, moving from a quickly fatal illness to a chronic disease [1]. Thus, although the initial therapeutic goal of HIV-infection management was patients’ survival, in the post-HAART era, the endpoint of both therapeutic approaches and research studies shifted to metabolic comorbidities and progressive population aging [2]. Among several chronic non-infectious comorbidities observed in HIV-infected patients together with increased life expectancy, the endocrine system is not spared from HIV infection [3]. Indeed, an impairment of the hypothalamic–pituitary–gonadal axis is frequently reported in HIV populations, although there is a highly variable incidence, depending on both the criteria used to diagnose hypogonadism and the clinical characteristics of patients enrolled [3]. Despite this heterogeneous picture, the well-known age-related decline in testosterone (Te) serum levels seems anticipated in HIV patients [4,5]. This progressive decline could be due either to a direct viral action on glandular tissue, or to the HAART-related sequelae, although the exact pathogenic mechanism remains to be elucidated [4,5]. Moreover, available studies assessing the prevalence of hypogonadism in HIV-infected patients reported very variable percentages, leaving open the question of the real interconnection between hypogonadism and HIV [3].

The relevance of hypogonadism in HIV settings has progressively increased in recent years, since it has been suggested as a marker of poor health status and frailty in the general population [6,7] and in HIV subjects [5,8]. Te and each related derivative compound (such as anabolic-androgenic steroids) [9] are used and, in terms of androgen anabolic properties, an involvement of Te reduction in HIV-related changes in body composition has been investigated [10]. Indeed, a variable degree of body composition alterations is described in HIV populations, such as subcutaneous lipoatrophy, increased amounts of visceral adipose tissue and wasting syndrome (WS), defined as an involuntary 10% weight loss in one year or a 5% weight loss in six months [11]. In general, HIV-related body composition changes and WS mainly concern the lean mass compartment, without an evident modification in total body weight [12]. This reduction could be measured by assessing the total amount of non-fat portions of the body, i.e., the lean body mass (LBM) (including a minimal quota of essential fat), the fat-free mass (FFM) (in which the essential fat is absent), and/or the body mass cell (BMC) (identifying all the metabolically active tissues of the body) [13]. The WS prevalence in HIV-infected patients is estimated at around 20% in the post-HAART period. Therefore, WS remains a relevant medical problem, considering its strict correlation with several metabolic comorbidities, reduced global performance status and impaired survival [12,14,15,16,17,18]. Since Te and/or Te-derived analogues are steroid molecules that are largely known to promote muscle mass gain [9,19,20], it is not surprising that exogenous androgenic treatment constituted an attractive therapeutic option for preventing WS in HIV populations. Hence, for many years, androgen treatments have been proposed in HIV settings, even in the absence of biochemical hypogonadism, for the contrasting of several disease-associated conditions, such as muscle mass lost, reduced bone mineral density, impaired mood and poor quality of life [21,22,23,24,25,26]. In 2005, a Cochrane meta-analysis evaluated the anabolic steroids efficacy on HIV-related WS, documenting a minimal increase in both LBM and body weight, despite several limitations due to the low number of included studies, the brief mean treatment duration and the heterogeneity of the subjects enrolled [20]. Similar results were thereafter reported by Corona et al. [27] in a further meta-analysis. However, other clinical trials were published over the years on this topic and no recent systematic literature review is still available regarding the effects of androgen administration on body composition in HIV-infected patients.

Considering the available evidence, this systematic review and meta-analysis was designed with a double aim: i) to better estimate the prevalence of hypogonadism in HIV subjects and the pituitary–testis hormonal profile in HIV patients in comparison to control men and ii) to comprehensively evaluate the effectiveness of Te and other androgenic compounds on body composition in HIV men.

## 2. Materials and Methods

We performed a meta-analysis according to the Cochrane Collaboration and PRISMA statement (Appendix A). To ensure originality and transparency of the review process, the meta-analysis was registered in the International Prospective Register of Systematic Reviews (PROSPERO; registration ID: CRD42018093170).

Two separate literature searches were performed. The first literature search was conducted to evaluate the prevalence of hypogonadism in HIV patients and to compare the pituitary–testis hormonal abnormalities between HIV-infected men and controls. The search was conducted until April 2021 for English-language articles published in the MEDLINE and Embase databases, using the following criteria: ((((((Testosterone) OR Androgen) AND HIV) OR Human Immunodeficiency virus) OR HIV man) OR HIV men) OR HIV infected men. The second literature search was performed to evaluate the effects of androgen administration on body composition in HIV-infected patients, using the following criteria: ((((((((((Testosterone) OR Testosterone administration) OR Testosterone therapy) OR Testosterone treatment) OR Androgen) OR Androgen administration) AND HIV) OR Human Immunodeficiency virus) OR HIV man) OR HIV men) OR HIV infected men.

### 2.1. Study Selection and Inclusion Criteria

#### 2.1.1. Literature Search 1

To identify the incidence of hypogonadism in HIV populations, the first literature search evaluated all clinical studies in which total Te (TT) serum levels were evaluated in HIV-infected men. In this first selection, both interventional and observational studies, either prospective or retrospective, were searched and collected, without specific inclusion or exclusion criteria for the study design. Randomization and presence of controls were not considered as inclusion criteria. Once longitudinal studies were considered, the enrolment visit was evaluated for data extraction.

Since the diagnosis of hypogonadism could be performed using different cut-offs [28,29], we selected those studies in which TT serum levels were reported, alongside the eventual percentage of hypogonadal patients calculated by authors. Thus, the inclusion criteria were:(i) HIV infection, (ii) male sex, and (iii) available TT serum levels. When both males and females were enrolled, the study was considered only if it was possible to discriminate hormonal data between the two genders.

#### 2.1.2. Literature Search 2

In order to evaluate the effects of androgen administration on body composition in HIV-infected men, the second literature search considered all randomized controlled clinical trials in which any kind of androgenic treatment was administered to HIV male patients. In particular, the inclusion criteria were: (i) HIV infection, (ii) male sex,(iii) randomized clinical trial, (iv) androgen administration, and (v) control group. All molecules with androgenic action were eligible and the analysis was performed separating Te and androgens-related molecules, such as nandrolone (N), oxandrolone and dehydroepiandrosterone (DHEA). Only studies evaluating the efficacy of androgen administration on body composition were included. Body composition was evaluated considering all possible parameters reported in each trial and only those parameters reported in at least three studies were meta-analysed. Each parameter was considered as the mean change after treatment in the study and control groups.

### 2.2. Data Collection Process and Quality

For both literature searches, three authors (DS, GS and WV) separately performed the selection, collecting abstracts of each study. All abstracts were evaluated for inclusion criteria and data were extracted from each study that was considered eligible, with regard to the study design, year of publication, number of included/excluded subjects and inclusion/exclusion criteria. AP, DS, GS, GC, MM, VR and WV performed quality control checks on the extracted data.

For the first literature search, the primary endpoint was TT serum levels in HIV-infected men. Secondary endpoints were: (i) prevalence of hypogonadism, (ii) Te cut-offs used to diagnose hypogonadism, (iii) fT serum levels, and (iv) sex hormone binding globulin (SHBG) serum levels.

For the second literature search, the primary endpoint was the body composition evaluation after androgen administration. Comparisons between HIV androgen-treated and HIV untreated men as well as between HIV androgens-treated subjects and healthy untreated controls were evaluated. Secondary endpoints were (i) Te serum levels and (ii) safety of androgen treatment, considering drop-out rates.

The risk of bias was assessed independently by three investigators (DS, GS and WV), using the Cochrane risk-of-bias algorithm. For each trial included in the comprehensive evaluation, several quality criteria and methodological details were considered. They included: (i) method of randomization, (ii) concealment of allocation, (iii) presence or absence of blinding to treatment allocation, (iv) duration and type of treatment and follow-up phases, (v) number of participants recruited, analysed or lost during the follow-up, (vi) timing of trial, (vii) whether an intention to treat analysis was noted, (viii) whether a power calculation was performed, (ix) source of funding, and (x) criteria for including participants and assessing outcomes.

### 2.3. Data Synthesis and Analysis

Using the Review Manager (RevMan) 5.3 software (Version 5.3.1, Copenhagen, Denmark, 2014), continuous variables were comprehensively evaluated as inverse variance of mean variables. Dichotomous variables were comprehensively evaluated using the Mantel–Haenszel method. Heterogeneity in mortality rate was assessed using I^2^ statistics. Even when a low heterogeneity was detected, a random-effects model was applied, because the validity of tests of heterogeneity can be limited with a small number of component studies. Funnel plots and the Begg-adjusted rank correlation test was used to estimate possible publication or disclosure bias. Sensitivity analyses were performed, considering studies in which androgens were administered and considering the presence at baseline of either hypogonadism or WS. In addition, a meta-regression analysis was performed to test the effect of different parameters on the prevalence of hypogonadism in HIV patients. Weighted mean differences and 95% CIs were estimated for the literature search. Values of *p* < 0.05 were considered statistically significant.

## 3. Results

### 3.1. Literature Search 1

In the first literature search, 15,631 papers were identified and 69 studies were selected for full text evaluation, since they fulfilled the inclusion criteria (Figure 1). Forty-six studies were included in qualitative and quantitative analyses. Among these studies, 30 were cohort and 16 case-control studies. The residual 23 studies were excluded after the full-text evaluation for unavailability of Te serum levels or because they were abstracts presented during congresses (Figure 1). Finally, 46 studies were evaluated and reported, and 41 studies were included in the analysis (Table 1 and Appendix A). In particular, 39 studies provided information on the prevalence hypogonadism, and 13 on TT levels, whereas data on SHBG and fT were available in five and six studies, respectively. The retrieved studies included 9466 and 654 HIV subjects under HAART treatment and controls, respectively. The mean age and body mass index (BMI) were 42.4 ± 6.2 years and 23.7 ± 1.9 kg/m^2^, respectively.

### 3.2. Hypogonadism Prevalence

The I^2^ value for the hypogonadism rate was 93.9, *p* < 0.0001. The funnel plot and Begg-adjusted rank correlation test suggested no major publication bias (Kendall’s τ: −0.03; *p* = 0.80). Overall, the prevalence of hypogonadism in HIV subjects was 26% [22,30] (Figure 2 and Appendix A, Panel A). The latter was significantly higher when fT was considered, as compared to TT (33% versus 19% *p* < 0.001). Those studies that used, as a cut-off for Te-deficiency, a TT value above 12 nmol/L were excluded from the analysis (Q = 8.93; *p* < 0.001, see also Figure 2 and Appendix A, Panel B), since they did not satisfy the criteria for the definition of hypogonadism [30]. Interestingly, when the whole population was considered, the prevalence of hypogonadism was inversely correlated with CD4^+^ cell count and year of publication (Appendix A, Panel A and B). Conversely, no association between hypogonadism rate and age, BMI or TT levels at enrolment was observed (not shown).

### 3.3. Hormonal profile 

The I^2^ value for TT concentration was 92.3, *p* < 0.0001. The funnel plot and Begg-adjusted rank correlation test suggested no major publication bias (Kendall’s τ: 0.14; *p* = 0.54). No significant difference for TT serum levels was observed between HIV patients and controls (Figure 3, Panel A). However, subjects with HIV showed significantly higher SHBG levels when compared to controls (Figure 3**,** Panel B). Accordingly, lower levels of fT were observed in HIV patients when compared to controls (Figure 3, Panel C). Data on gonadotropin levels were available in five studies (Table 1). No significant differences between cases and controls for FSH levels were observed (Figure 3, Panel D), whereas a trend toward higher LH levels in HIV subjects was observed (*p* = 0.07, Figure 3, Panel E).

### 3.4. Literature Search 2

The second literature search identified 23,636 papers, and 33 studies were included in the full-text analysis, since they fulfilled the inclusion criteria (Figure 1). After the full-text evaluation, 17 studies were finally considered (Figure 1 and Table 2). Among the 12 trials excluded during the full text evaluation, four trials did not contemplate a control group, four trials did not report body composition parameters, seven trials considered male and female patients together and one provided a previous Te administration in both groups before androgen supplementation (Appendix A) (Figure 1).

Finally, 1267 HAART-treated HIV-infected men were considered in the analysis, 661 treated with androgens and 606 with placebo. Two sensitivity analyses were performed considering patients’ baseline characteristics. First, papers evaluating the use of Te were divided according to the presence of hypogonadism (eight studies) considering the cut off of 12.1 nmol/L [31,32,33,34,35,36,37,38] or the absence of hypogonadism (eight studies) [10,21,25,39,40,41,42,43] as an inclusion criterion. Second, papers evaluating the use of Te were divided according to the presence (13 studies) [10,21,32,33,34,35,36,39,40,41,42,43] or absence (four studies) [25,31,37,38] of WS as an inclusion criterion.

### 3.5. Efficacy—Testosterone Levels

Only 11 out of 17 trials reported changes in TT serum levels after treatment, for a total of 632 patients. Although the overall Te change was not significant after treatment (5.80 nmol/L: −0.53; 12.14, *p* = 0.070), as expected, a significant increase in Te was observed when Te was administered (11.61 nmol/L: 6.09; 17.12, *p* < 0.001) but was not observed when androgen-related molecules were used (−8.20 nmol/L: −18.00; 1.60, *p* = 0.010) (Appendix A Panel A). Sensitivity analyses confirmed this significant increase in both hypogonadal (11.47 nmol/L: 1.19; 21.75, *p* = 0.030) and not-hypogonadal men (13.20 nmol/L: 12.80; 13.60, *p* < 0.001) (Appendix A Panel B). Moreover, the Te administration significantly changed TT serum levels in men with WS (12.99 nmol/L: 7.37; 18.62, *p* < 0.001) (Appendix A Panel C).

Androgen administration did not globally influence fT serum levels (19.02 pmol/L: −3.54; 41.57, *p* = 0.100), although a significant increase was observed after the use of Te (16.11 pmol/L: 8.64; 23.58, *p* < 0.001) rather than androgen-related molecules (Appendix A). 

Changes in SHBG were reported in 11 studies, accounting for 343 patients. The administration of androgens significantly reduced the SHBG serum levels (*p* < 0.001) (Figure 4 Panel A). This reduction was confirmed after separately considering Te and other androgen-related molecules (Figure 4 Panel A). The difference in SHBG occurred at the limit of statistical significance in hypogonadal men (−13.26: −26.67; 0.15, *p* = 0.050), whereas only one study considered HIV-infected men without hypogonadism at baseline (Figure 4 Panel B). The reduction in SHBG serum levels was confirmed in men treated with WS (−15.92: −24.30; −7.54, *p* < 0.001), while only one study evaluated HIV-infected men without WS (Figure 4 Panel C).

### 3.6. Body Composition

Body composition parameters were assessed by dual energy x-ray absorptiometry (12 studies) or by bioelectrical impedance analysis (five studies). 

Body weight changes (WC) were reported in 12 studies, for a total of 734 patients. WC was lower when a placebo was administered instead of androgen therapy, although the difference did not reach a statistically significant level (*p* = 0.060) (Figure 5, Panel A). However, in 11 trials, Te was dispensed, whereas only one study used nandrolone (N). Considering only the action of Te, and comparing treated and untreated patients, the WC increase was significantly higher in the study group than in the control group (*p* = 0.008) (Figure 5, Panel A). Sensitivity analyses showed that body weight significantly changed when Te was administered in hypogonadal men (1.27 kg: 0.30; 2.24, *p* = 0.010), but not in eugonadal HIV-infected men (0.59 kg: −0.66; 1.85, *p* = 0.360) (Appendix A Panel A). Considering the presence or absence of WS as an inclusion criterion, the administration of T significantly increased body weight in men with WS (1.13 kg: 0.39; 1.86, *p* = 0.003) (Appendix A Panel B), whereas only one study considered HIV-infected men without WS. 

The LBM change was reported in eight studies for a total of 447 patients and it was higher in the study group compared to the control group (*p* < 0.001) (Figure 5, Panel B). This androgen-mediated increase in LBM remains evident when also considering Te and N administration separately, although this effect seems to be more pronounced when using N instead of Te. Sensitivity analyses showed that the change in LBM was not significantly different when Te was administered in hypogonadal (0.97: −0.43; 2.36, *p* = 0.170) and not-hypogonadal men (1.70: −1.63; 5.04, *p* = 0.320) (Appendix A Panel A). Since only one study considered HIV-infected men without WS, the comprehensive evaluation of Te administration in WS patients revealed no significant changes in LBM (1.20: −0.59; 2.99, *p* = 0.190) (Appendix A Panel A). Thus, sensitivity analyses suggested that the androgen effect on LBM could not be predicted by hypogonadism or WS occurrence.

Changes in FFM were reported in six studies for a total of 251 patients, with higher results in the study group than in the control group (*p* < 0.001) (Figure 5, Panel C). This significant difference also remained when dividing studies between those using Te (four studies) and those using N (two studies) (Figure 5, Panel C). Sensitivity analyses were not possible for evaluation of the presence of hypogonadism at baseline, since changes in FFM were not reported in studies evaluating not-hypogonadal men. Considering HIV-patients with or without WS at baseline, only one study evaluated men without WS and the remaining group of studies showed a significant change in FFM (2.82: 2.00; 3.63, *p* < 0.001). Similarly to LBM, these results suggest that changes in FFM could not be predicted by hypogonadism or WS at baseline.

Changes in fat mass (FM) were reported in nine studies, including a total of 537 patients. This change was not significantly different comparing treated and untreated HIV-positive men (0.03: −0.41; 0.47, *p* = 0.900) (Appendix A Panel A), independently of the androgenic compound used. However, sensitivity analyses showed that androgen administration significantly affected the FM in both hypogonadal (*p* = 0.040) and not-hypogonadal men (*p* = 0.020), although with opposite directions of causality (Appendix A Panel B). On the other hand, sensitivity analysis was not possible for the presence of WS, since no studies evaluating HIV-infected men without WS were available.

Changes in BMC were reported in five studies for a total of 475 patients. The administration of androgens did not change the BMC (0.47: −0.26; 1.20, *p* = 0.210), independently of the molecule used. Sensitivity analyses were not possible considering the low number of studies included in this subgroup analysis.

### 3.7. Safety

All studies included in the analysis reported the dropout rate. The odds ratio of the dropout rate did not significantly differ when comparing the study and control groups (Figure 6).

## 4. Discussion

Our systematic review and meta-analysis found that the prevalence of hypogonadism in HIV subjects is particularly high, especially when fT was considered. Accordingly, SHBG is increased and fT reduced in HIV subjects when compared to controls. In line with the latter results, a trend toward higher LH levels in HIV-infected men was observed, especially in older studies. Interventional analysis depicts a clear effect of androgen administration in HIV-infected men at increasing body weights, mainly due to a significant change in the lean body compartment.

The overall prevalence of hypogonadism in HIV-infected men was around 26% and up to 40% when fT was considered, confirming previous results [3]. Several narrative reviews attempted to describe the prevalence of hypogonadism in this group of patients, highlighting the high heterogeneity of the studies and, overall, of the methodologies used to measure and define male hypogonadism [3]. Present data show that the prevalence of hypogonadism in HIV subjects is similar to that reported in other subjects with chronic diseases [44], such as those with diabetes mellitus [45], metabolic syndrome (MetS) [46], or erectile dysfunction [47], or who are using drugs that potentially interfere with T production [48,49]. Conversely, HIV patients show from 10 to 20 times higher prevalence of hypogonadism when compared to the general population [50]. In addition, this meta-analysis confirms the results of a few studies showing that hypogonadism occurs earlier in HIV-infected men [4] when compared HIV-uninfected men since age was not associated with the occurrence of hypogonadism, as is generally expected in the general population [50]. A combination of genetic, environmental and age-related factors is supposed to play the major role in determining the development of LOH [51,52]. The specific mechanisms underlying the development of hypogonadism in HIV patients are still largely unknown. Present data show similar TT serum levels between HIV-infected and uninfected men, when compared in the same setting. However, subjects with HIV showed higher SHBG and lower fT serum levels compared to healthy controls. In addition, no difference in gonadotropin levels was observed although a trend toward higher LH levels was found, suggesting the presence of mixed hypogonadism. It should be considered, however, that the trend for higher serum LH found in our meta-analysis refers mainly to studies performed in the 1990s (Figure 3 Panel D), including HIV patients who had survived the disease in the pre-HAART era; these patients were exposed to several opportunistic infections that, at the time, often also involved the testis, thus accounting for primary hypogonadism [53,54]. A large body of evidence has suggested that overall poor health status represents a major risk factor in HIV-related hypogonadism [5,8,12]. In line with the latter observation, we observed an inverse relationship between CD4+ cell count and the HIV-related prevalence of hypogonadism, suggesting that a worse HIV-infection control increases the risk of hypogonadism onset. Accordingly, hypogonadism was more frequently detected in the pre-ART area when opportunistic infections and overall poor health status were more frequently observed [3,12]. In line with the latter hypothesis, we reported an inverse relationship between HIV hypogonadism and the year of study publication. As was also the case for for other HIV comorbidities, it seems that the prevalence of hypogonadism tended to decrease in recent years due to better control of HIV infection and the minor levels drug toxicity provided by newer antiretroviral drugs used alone or in combination [55]. Other factors that can negatively interfere with Te production in HIV patients include metabolic derangement as well as concomitant liver alterations with Hepatitis B Virus (HBV) or HCV co-infection, and the presence of inflammatory cytokines such as TNF-α and interleukin-1 [12]. Finally, ART per se, as well as the concomitant use of other drugs such as glucocorticoids, megestrol acetate, psychotropic medications and opiates or methadone, can all contribute to the development of hypogonadism in HIV populations [3,12].

When interventional studies were investigated, body weight modifications were limited to the use of T in hypogonadal men or HIV patients with WS. In addition, we found that body composition was changing, when considering either LBM or FFM, independently of the androgenic compound administered. Although these parameters are both expressions of the lean mass compartment and differ only in terms of the presence or absence of the essential fat, the administration of androgens seems not to exert a comparable effect. In particular, the action of androgens on LBM seems to not be influenced by baseline differences in terms of the presence of hypogonadism and/or WS. On the contrary, considering FFM, these baseline conditions seem to affect the action of androgens in HIV-infected men, which is modified in hypogonadal HIV subjects treated with Te, but not in eugonadal ones. The beneficial action of androgens on FFM is well known and was previously reported in other meta-analyses performed in HIV-uninfected men, in particular in middle-aged men [56], in patients with MetS [46] or in Klinefelter syndrome [57]. Although HIV patients present peculiar disease- and treatment-associated comorbidities, compared to the uninfected population, ageing males could be partially comparable, since they share some degree of metabolic alteration, including body composition changes. However, measuring the effect of chronic androgen administration on body composition remains challenging, as demonstrated by the lack of significant changes of BMC. Indeed, although the depletion of BMC represents one of the most appropriate criteria to identify the presence of WS [58], the evaluation of the effects of androgens on this outcome remains poor, probably due to the low number of studies reporting this parameter. Moreover, LBM, FFM and BMC could be measured by a significantly high number of diagnostic procedures, which could bias the final results due to heterogeneity. Thus, a single lean mass-related parameter and a specific diagnostic tool for this parameter remain far from being universally validated.

The action of androgens on the lean body compartment is independent of the molecule used, since changes are significant using both Te and N. However, only one study compared these two androgenic compounds in the same clinical setting [10], providing a direct comparison between the two arms and supporting a superior therapeutic effect of N in HIV-infected men with WS. These molecules differ in terms of the presence (Te) or the absence (N) of the methyl group in position C_19_, supplying N with enhanced anabolic effects and reduced androgenic properties [59]. Although the available data are not sufficient to highlight the predominance of a specific androgenic molecule in terms of the improvement of body composition parameters in HIV patients, it is important to emphasize that preliminary data suggest that oestrogens play a limited role on lean mass, which is mainly under the control of Te in men [60]. Moreover, the increasing use of N as a doping substance [61] has limited its clinical application, reducing the availability of data on this compound in the setting of HIV-infected patients.

Despite the demonstration of the efficacy of androgens in terms of body composition in HIV-infected men, the real mechanism through which this action is carried out remains unclear. Indeed, whether this beneficial change is mediated by the modification of serum Te levels is still debated. In the present meta-analysis, we could only evaluate an association between body changes and therapy, but not a cause-and-effect relationship. As expected, androgen administration increases TT serum levels only when Te is used instead of Te-analogues, independently of the gonadal baseline status. Accordingly, the same result is observed when considering fT levels. However, all androgen administrations reduced SHBG with a potential final increase in fT serum levels. Thus, despite these significant variations in serum hormonal levels, the biochemical mechanism by which Te affects body composition is not completely understood. The androgens’ capability to increase LBM, and consequently muscle mass, has been previously envisaged in different experimental settings [62,63]. In fact, Te is able to stimulate lipolysis in animal models [64], as well as in human subcutaneous adipose tissue, in which both a Te-induced inhibition of lipoprotein lipase and an activation of the hormone sensitive lipase were observed [65]. Physiological and supra-physiological Te levels were demonstrated to increase the number and the size of muscle fibres by stimulating protein synthesis [66]. The in vitro evaluation of the myoblasts’ culture system highlighted the capability of Te to induce muscle hypertrophy, as well as a non-genomic action, by activating a G-protein-linked receptor, which resulted in cellular growth [67]. With this in mind, it is not surprising that androgen treatment induces changes in muscle tissue, without detectable effects on adipose tissue, as confirmed by the absence of overall modifications in FM. However, a significant increase in FM was observed in hypogonadal Te-treated patients, while a significant decrease was observed in eugonadal HIV patients. The limited number of trials evaluating FM does not allow the obtaining of conclusive results, although we could speculate that the restoration of eugonadism in androgen-deficient patients could indirectly impact FM by improving general health status, mood and appetite. On the contrary, supra-physiological Te levels could act predominantly on the lean mass compartment, with a relative reduction in FM at equal body weight.

Sensitivity analyses show that the presence or absence of hypogonadism as well as of WS are not useful criteria for identifying patients who could benefit from androgenic treatment in terms of body composition. However, these subgroup analyses should be considered with caution, since the statistical power is reduced due to the limited number of included trials. Moreover, the vast majority of included studies evaluated patients with a history of WS or body weight reduction. These conditions, although typical of HIV infection, are heterogeneously defined [68], and this could be another relevant bias in the interpretation of sensitivity analyses.

Finally, all the results reported in the present meta-analysis should be cautiously evaluated, considering the high degree of heterogeneity among included studies. This variability is due to the absence of shared evidence-based parameters that are able to measure the lean body compartment, and to the high variability among studies in terms of the inclusion criteria, the mode of measurement of body composition, the androgenic compounds selected, dosages, and scheme and route of administration. In addition, while only a few trials were designed to assess the lean mass compartment change in HIV settings, thus limiting data availability, the androgen therapeutic schemes were offered for a maximum of six months, making long-term considerations impossible. All these parameters should be considered, together with the low quality of the included studies, as depicted in the risk of bias evaluation, suggesting the high levels of difficulty involved in designing a clinical trial in HIV-infected patients.

In conclusion, our systematic review highlights an increased prevalence of hypogonadism in the young to middle-aged HIV-population compared to uninfected men. However, this comorbidity could remain neglected when only TT serum levels are measured. Interestingly, reduced Te serum levels reflect poor disease control, which was related to CD4^+^ cell count. In HIV-infected men, chronic androgen supplementations seem to exert a significant impact on body composition, particularly affecting the lean mass compartment. Indeed, the androgenic treatment for anabolic purposes in HIV male patients leads to a mild improvement of some body composition parameters, although this is not predictable. Moreover, since the vast majority of trials set up short-term treatment schemes, no data are currently available regarding the long-term effects of androgen administration in a HIV setting. Considering the general frailty of HIV patients and the significant burden of adverse effects that characterizes anabolic androgenic therapy, a personalized and tailored therapeutic approach is mandatory in such patients.

## Figures and Tables

**Figure 1 cells-10-02067-f001:**
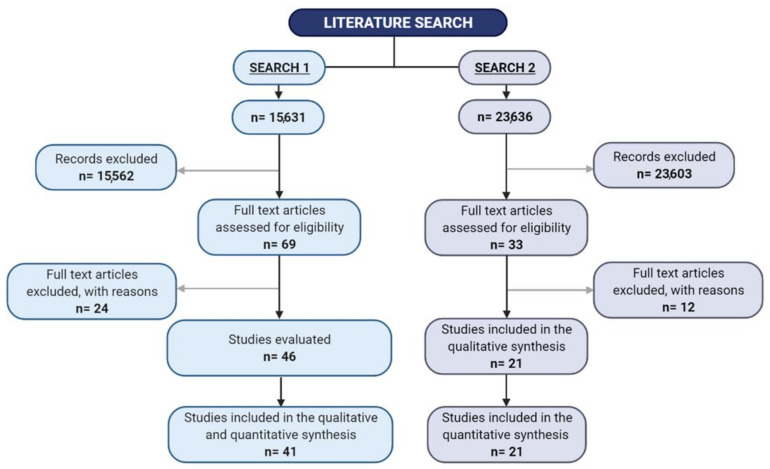
Study flow chart.

**Figure 2 cells-10-02067-f002:**
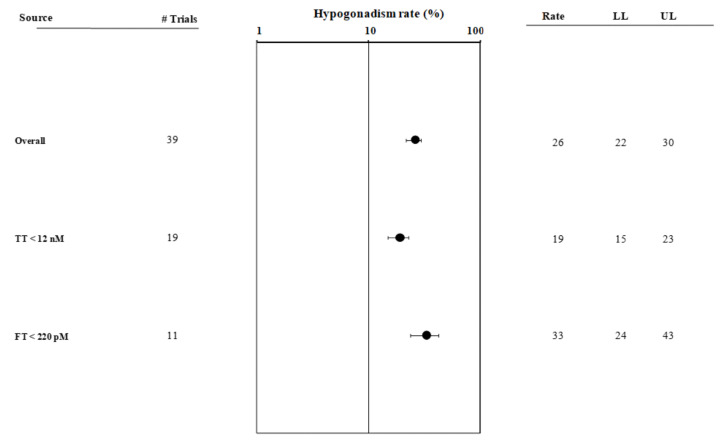
Forest plot showing hypogonadism prevalence in studies detected in the first literature search. fT: free testosterone; LL: lower limit; UL: upper limit; TT: total testosterone.

**Figure 3 cells-10-02067-f003:**
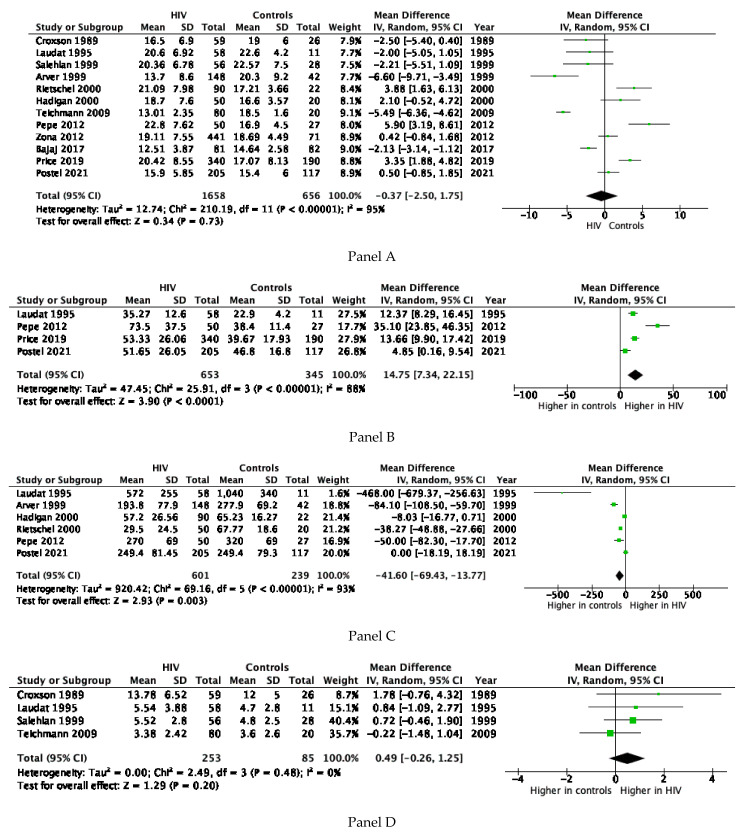
Forest plot comparing HIV-infected patients and controls detected in the first literature search for total testosterone serum levels (panel **A**), sex hormone binding globulin (SHBG) (panel **B**), free testosterone (Panel **C**), follicle stimulating hormone (FSH) (panel **D**), and luteinising hormone (LH) (panel **E**).

**Figure 4 cells-10-02067-f004:**
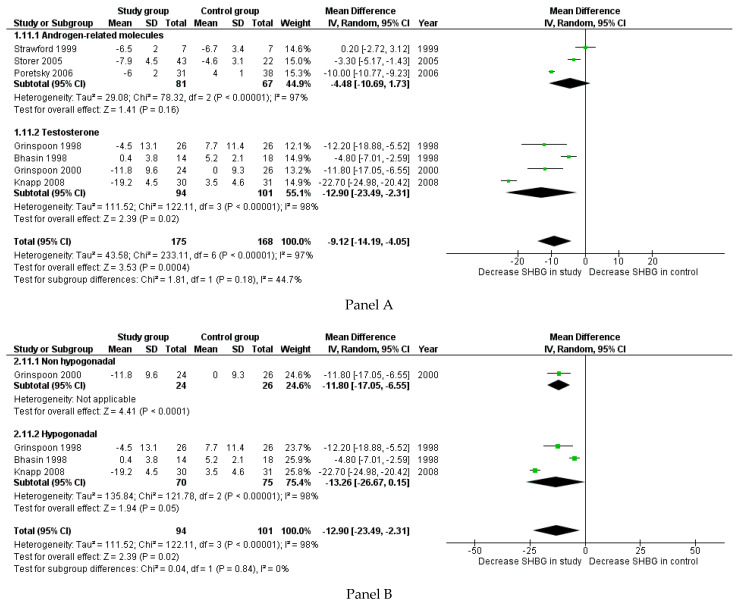
Forest plot reporting groups detected in the second literature search for changes in sex hormone binding globulin (SHBG) between study and control groups (Panel **A**). Panel **B** reports the sensitivity analysis for hypogonadism, and Panel **C** shows the sensitivity analysis for wasting syndrome.

**Figure 5 cells-10-02067-f005:**
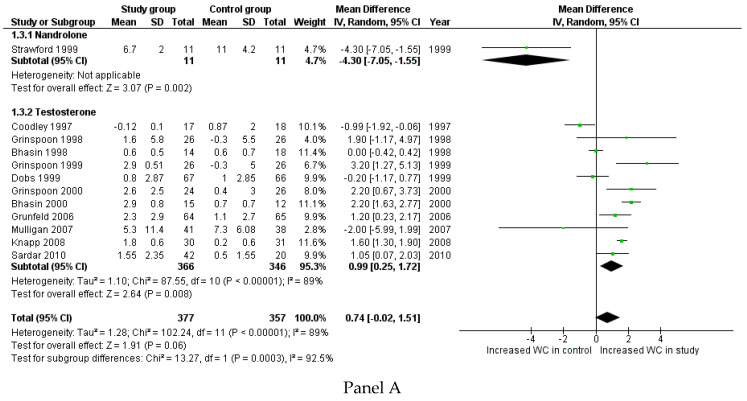
Forest plot comparing study and control groups detected in the second literature search, considering body weight change (WC) (kg) (Panel **A**), lean body mass (LBM) (Panel **B**), and fat free mass (FFM) change (Panel **C**).

**Figure 6 cells-10-02067-f006:**
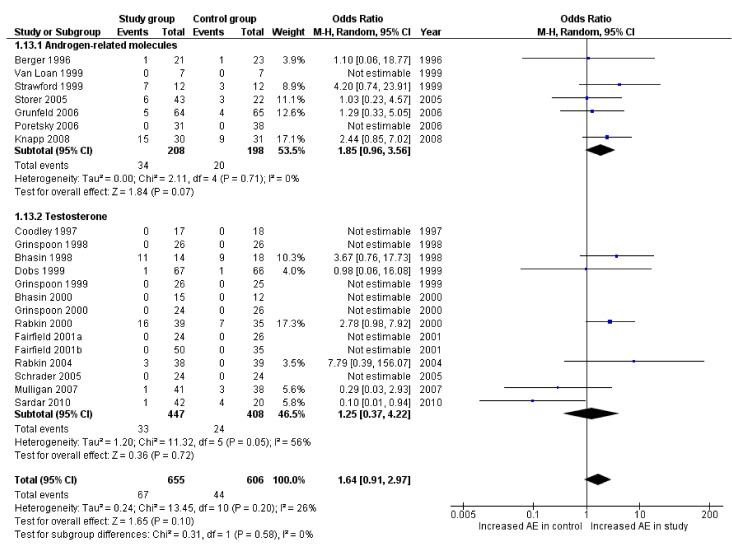
Forest plot reporting drop out occurrence between study and control groups detected in the second literature search.

**Table 1 cells-10-02067-t001:** Characteristics of studies detected with the first literature search.

Author Name	Year	Study Design	Definition of Hypogonadism	Control Group	Control Group Criteria	HIV Patients (n)	HIV Patients Age (Years)	Controls (n)	Controls Age (Years)
Dobs	1988	One-arm, Longitudinal	NA	No	NA	19	35 ± 13.1		
Croxson	1989	Longitudinal	TT < 19 nmol/L	Yes	Healthy homosexual	32	32.9 ± 4.7	26	32.9 ± 4.7
Raffi F	1991	Cross-sectional	TT < 10.4 nmol/L	No	NA	67			
Laudat	1995	Cross-sectional	NA	Yes	Healthy	58	36.1 ± 8.6	11	36.8 ± 5.5
Grinspoon	1996	Cross-sectional	fT < 41.6 pmol/L	No	NA	77	39 ± 7		
Dobs	1996	Cross-Sectional	NA	No	NA	13	45 ± 7.2	13	44 ± 6.4
Bhasin	1998	RCT	NA	No	NA	32			
Salehian	1999	Longitudinal	TT < 12.14 nmol/L	Yes	Healthy	56	34 ± 6	28	32 ± 7
Arver	1999	Longitudinal	TT < 9.5 nmol/L	Yes	Healthy	148	40 ± 1	42	31 ± 2
Kopicko	1999	Cross-sectional	TT < 13.9 nmol/L	No	NA	587	38.3 ± 8		
Dobs	1999	RCT	TT < 13.9 nmol/L	No	NA	123	40 ± 7.5		
Rietschel	2000	Cross-Sectional	NA	Yes	Healthy	90	39 ± 1	22	38 ± 1
Hadigan	2000	RCT	NA	Yes	Healthy	50	38.9 ± 6.4	20	35.2 ± 6.3
Grinspoon	2000	RCT	fT < 41.6 pmol/L	No	NA	61	41.4 ± 2		
Rabkin	2000	RCT	TT < 10.4 nmol/L	No	NA	70	39.1 ± 8.1		
Biglia	2004	case-control	NA	No	NA	84	43.5 ± 10.5		
Crum-Cianflone	2007	Cross-Sectional	TT < 10.4 nmol/L	No	NA	300	39.4 ± 9.2		
Andersen	2007	Cross-Sectional	NA	No	NA	16	50 ± 8		
Mulligan	2007	RCT	NA	No	NA	79	40.1 ± 10.6		
Knapp	2008	open-label switch study	TT < 10.4 nmol/L	No	NA	61	43.2 ± 6.7		
Teichmann	2009	Cross-sectional	TT < 10.4 nmol/L	Yes	Healthy	80	37.1 ± 5.7	20	35.4 ± 4.1
Moreno-Pérez	2010	Cross-sectional	fT < 220 pmol/L	No	NA	90	42 ± 8.2		
Rochira	2011	Cross-sectional	TT < 10.4 nmol/L	No	NA	1325	44.74		
Pepe	2012	Cross-sectional	fT < 225 pmol/L	Yes	Healthy	50	48.6 ± 9.4	27	49.1 ± 8.3
Zona S	2012	Cross-sectional	TT <10.4 nmol/L	Yes	Healthy	441	44.8 ± 5.9	71	36.7 ± 11.9
Sunchatawirul	2012	Cross-sectional	fT < 225 pmol/L	No	NA	491	37.1 ± 1.7		
Guaraldi	2012	Cross-sectional	TT < 10.4 nmol/L	no	NA	133	48.8 ± 7.4		
De Ryck	2013	Cross-sectional	fT < 220 pmol/L	No	NA	49	48.7 ± 2.9		
Pérez	2013	Cross-sectional	TT < 10.4 nmol/L	No	NA	158	45.8 ± 4.1		
Pepe	2014	Cross-sectional	fT < 225 pmol/L	No	NA	41	48.3 ± 8.3		
Rochira	2015	Cross-sectional	TT < 10.4 nmol/L	No	NA	1359	45 ± 1.2		
Bhatia	2015	Cross-sectional	NA	No	NA	992	44 ± 2		
Pathak	2015	Longitudinal	NA	No	NA	45	36.8 ± 8.8		
Gomes	2016	Cross-sectional	TT < 9.7 nmol/L or fT < 83.7 pmol/L	No	NA	245	48 ± 11.2		
Santi	2016	Cross-sectional	TT < 10.4 nmol/L	No	NA	1204	45.6 ± 7.3		
Bajaj	2017	Cross-sectional	TT < 10.4 nmol/L	Yes	Healthy	81		82	
Dutta	2017	Cross-sectional	TT < 10.4 nmol/L	No	NA	225	39.5 ± 41		
Price	2019	Cross-sectional	NA	Yes	Healthy	340	52.0 ± 1.5	190	54.1 ± 2.2
Bajaj	2020	Cross-sectional	TT < 8.36 nmol/L	No	NA	84	42.3 ± 10.5		
Pezzaioli	2020	Cross-sectional	TT < 12 nmol/L or fT < 83.7 pmol/L	No	NA	94	53.1 ± 1.8		
Postel	2021	Cross-sectional	fT < 225 pmol/L	Yes	Healthy	205	61.5 ± 7.2	117	62 ± 8.1
de Vincentis	2021	Cross-sectional	TT < 10.4 nmol/L	no	NA	316	45.3 ± 5.3		
Quiros-Roldan	2021	Cross-sectional	TT < 12 nmol/L or fT < 83.7 pmol/L	No	NA	107	53.9 ± 2.0		
Pilatz	2021	Longitudinal	TT < 8 nmol/L	No	NA	87	43.1 ± 2.2		

fT: free testosterone; HIV: human immunodeficiency virus; NA: not available; RCT: randomized clinical trial; TT: total testosterone.

**Table 2 cells-10-02067-t002:** Characteristics of studies included in the meta-analysis (search 2).

Author	Year	Androgen Molecule(s)	Androgen Dosages	Treatment Duration	Endpoints	Study Group—Type of Patients	Control Group—Type of Patients
Berger JR	1996	Oxandrolone	5 or 15 mg/day	16 weeks	Body composition and QoL	HIV-associated weight loss	HIV-associated weight loss
Coodley GO	1997	T cypionate	200 mg biweekly	12 weeks	Body composition	HIV-associated weight loss	HIV-associated weight loss
Bhasin S	1998	T	5 mg/day	12 weeks	Body composition and QoL	HIV-hypogonadal (TT < 13.88 nmol/l)	HIV-hypogonadal (TT < 13.88 nmol/l)
Grinspoon S	1998	T enanthate	300 mg triweekly	24 weeks	Body composition and QoL	HIV-associated weight loss and low Te	HIV-associated weight loss and low Te
Strawford A	1999	Nandrolone decanoate	65 or 200 mg per week	12 weeks	Body composition and metabolic parameters	HIV-infected men with WS	HIV-infected men with WS
Van Loan MD	1999	Nandrolone decanoate	65 or 195 mg per week	3 weeks	Body composition	HIV men weight loss >5% body weight, serum Te levels <25th percentile for age-group or <33rd percentile with hypogonadal symptoms	HIV men weight loss >5% body weight, serum Te levels <25th percentile for age-group or <33rd percentile with hypogonadal symptoms
Dobs AS	1999	T	6 mg/day	12 weeks	Body composition and QoL	HIV-associated weight loss (5–20%) and low Te (<13.88 nmol/L)	HIV-associated weight loss (5–20%) and low Te (<13.88 nmol/L)
Grinspoon S	1999	T enanthate	300 mg triweekly	24 weeks	Body composition	HIV-associated weight loss and low Te	HIV-associated weight loss and low Te
Bhasin S	2000	T enanthate	100 mg weekly	16 weeks	Body composition	HIV-associated weight loss (>5%) and low Te (<12.1 nmol/L)	HIV-associated weight loss (>5%) and low Te (<12.1 nmol/L)
Grinspoon S	2000	T enanthate	200 mg weekly	12 weeks	Body composition	HIV-infected men and WS (weight <90% IBW)	HIV-infected men and WS (weight <90% IBW)
Fairfield WP	2001	T enanthate	200 mg weekly	12 weeks	Body composition	HIV-infected men with WS (weight <90% of IBW or weight loss >10% of baseline weight)	HIV-infected men with WS (weight <90% of IBW or weight loss >10% of baseline weight)
Storer TW	2005	Nandrolone decanoate	150 mg biweekly	12 weeks	Body composition	HIV-associated weight loss (5–15% over 6 months)	HIV-associated weight loss (5–15% over 6 months)
Schrader S	2005	T	5 g	4 weeks	Te efficacy	HIV-hypogonadal	HIV-hypogonadal
Poretsky L	2006	DHEA	100 mg to 400 mg orally daily	8 weeks	Endocrine and metabolic parameters	HIV-infected men with mild depression defined by 3< x >5 criteria of DSM IV	HIV-infected men with mild depression defined by 3< x >5 criteria of DSM IV
Grunfeld C	2006	Oxandrolone	20 or 40 or 80 mg orally daily	12 weeks	Body composition	HIV-infected men with WS syndrome (10%–20% weight loss or BMI <20 kg/m^2^)	HIV-infected men with WS syndrome (10%–20% weight loss or BMI <20 kg/m^2^)
Mulligan K	2007	Megestrol ± Testosterone	800 mg daily ± 200 mg biweekly	12 weeks	Body composition	HIV-positive men with 5% or more weight loss or BMI <20 kg/m^2^	HIV-positive men with 5% or more weight loss or BMI <20 kg/m^2^
Sardar P	2010	T and Nandrolone	250 and 150 mg biweekly	12 weeks	Testosterone vs Nandrolone	HIV-infected men with WS	HIV-infected men with WS

AIDS: acquired immunodeficiency syndrome; BMD: bone mineral density; BMI: body mass index; DHEA: dehydroepiandrosterone; DSM: diagnostic and statistical manual of mental disorders; ED: erectile dysfunction; HIV: human immunodeficiency virus; IBW: ideal body weight; QoL: quality of life; Te: testosterone; WS: wasting syndrome.

## Data Availability

Data supporting reported results could be obtained after direct author request.

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
