# Peer review of "The Prevalence of Hypogonadism and the Effectiveness of Androgen Administration on Body Composition in HIV-Infected Men: A Meta-Analysis"

_cells, 2021, doi:10.3390/cells10082067_

Round 1

Reviewer 1 Report

This systematic review is interesting and well written. I think that the authors have tried to make a complete approach to the complex entity that is the hypogonadism in HIV-infected patients. I still have several reservations that need clarification.

  • Lines 56-58: This sentence is not clear. I think that the authors should cite the major studies assessing prevalence of hypogonadism in HIV-infected patients and give the range of prevalence reported in order to demonstrate that there is variability in current data

  • Results, Chapter 3.2, lines 200-205. I have some difficulty understanding the results on the prevalence of hypogonadism (26%). Was that based on the criteria defined by each individual study? If yes and if this is a categorical variable (presence or absent), how do the authors report a range (22, 30)? In the same section what were the cut-off to define hypogonadism based on TT and free testosterone? < 12 nmol/l and < 220 pmol/l respectively? I think that these points should be better clarified in Methods

  • Lines 246-252: In Figure 1, the authors state that they included 21 studies for the second literature review. However, in the sensitivity analysis, only 17 studies were included. I think that the authors should complete Figure 1 to clarify how many studies were included in the sensitivity analysis in each arm and why the other studies were not included. 

  • Figure 5: I consider confusing the fact that LBM data (panel B) are shown as absolute values and not as change scores, as it is the case for total weight and FFM. I think that the Figure should be reworked to be more homogeneous

  • Lines 332-335: This conclusion is unsupported by the data. There were no studies assessing FFM changes in non-hypogonadal men as noted in lines 328-330. Thus, I do not understand how the authors conclude that FFM change could be predicted by hypogonadism status at baseline...

  • Lines 432-434: I think that the authors should revise this sentence. Nadrolone is less aromatized than testosterone, so there is no increased possibility of nadrolone to be aromatized as it is stated in this part.

  • It would be useful that the authors detail better the method by which the body composition was assessed. How many of these studies used DXA, which is considered the gold standard for whole body – composition?

  • I think that the authors should even more stress the fact that the main limitation is the limited number of studies available for body composition (only 5-6 for LBM and FFM). They should also acknowledge more clearly in the limitation that there are only short-term studies (maximum duration 6 months) and that some of the differences could not be statistical significant because of this fact.

Author Response

Reviewer 1

This systematic review is interesting and well written. I think that the authors have tried to make a complete approach to the complex entity that is the hypogonadism in HIV-infected patients. I still have several reservations that need clarification.

1) Lines 56-58: This sentence is not clear. I think that the authors should cite the major studies assessing prevalence of hypogonadism in HIV-infected patients and give the range of prevalence reported in order to demonstrate that there is variability in current data

ANSWER: Thank you for your comment. We rephrased the sentence according to your suggestion as follows: ‘Moreover, available studies assessing the prevalence of hypogonadism in HIV-infected patients reported very variable percentages, leaving open the question of the real interconnection between hypogonadism and HIV’.

2) Results, Chapter 3.2, lines 200-205. I have some difficulty understanding the results on the prevalence of hypogonadism (26%). Was that based on the criteria defined by each individual study? If yes and if this is a categorical variable (presence or absent), how do the authors report a range (22, 30)? In the same section what were the cut-off to define hypogonadism based on TT and free testosterone? < 12 nmol/l and < 220 pmol/l respectively? I think that these points should be better clarified in Methods

ANSWER: The hypogonadism prevalence was obtained considering the definition suggested in each study included in the analysis. The definition of hypogonadism followed the thresholds suggested by each author. We added this information within the text.

3) Lines 246-252: In Figure 1, the authors state that they included 21 studies for the second literature review. However, in the sensitivity analysis, only 17 studies were included. I think that the authors should complete Figure 1 to clarify how many studies were included in the sensitivity analysis in each arm and why the other studies were not included. 

ANSWER: Thank you for this clarification. Finally, only 17 studies were included in the second literature review since four did not report body composition parameters as outcomes. We corrected the text and Figure 1 accordingly.

4) Figure 5: I consider confusing the fact that LBM data (panel B) are shown as absolute values and not as change scores, as it is the case for total weight and FFM. I think that the Figure should be reworked to be more homogeneous

ANSWER: Thank you for your observation. However, we analysed data as reported in studies included. Indeed, weight and FFM were reported as both absolute values and change. Thus, we used the change data for the analyses. On the contrary, LBM data were available only as absolute value. Thus, we have to maintain this kind of analysis.

5) Lines 332-335: This conclusion is unsupported by the data. There were no studies assessing FFM changes in non-hypogonadal men as noted in lines 328-330. Thus, I do not understand how the authors conclude that FFM change could be predicted by hypogonadism status at baseline...

ANSWER: We agree with the reviewer’s comment. We corrected the sentence as follows: ‘Similarly to LBM, these results suggest that FFM change could not be predicted by hypogonadism or WS at baseline’.

6) Lines 432-434: I think that the authors should revise this sentence. Nandrolone is less aromatized than testosterone, so there is no increased possibility of nandrolone to be aromatized as it is stated in this part.

ANSWER: We agree with the reviewer’s comment. We corrected the sentence as follows: ‘…supplying to N enhanced anabolic effects and reduced androgenic properties’.

7) It would be useful that the authors detail better the method by which the body composition was assessed. How many of these studies used DXA, which is considered the gold standard for whole body – composition?

ANSWER: Thank you for this comment. Considering the 17 studies included, 12 measured body composition parameters through DXA, while other five used bioelectrical impedance analysis. We added this information in the results section as follows: ‘Body composition parameters were assessed by dual energy x-ray absorptiometry (12 studies) or by bioelectrical impedance analysis (5 studies)’. Moreover, we added this limit in the discussion section as follows: ‘Finally, all the results reported in the present meta-analysis should be cautiously evaluated, considering the high degree of heterogeneity among studies included. This variability is due to the absence of shared evidence-based parameters able to measure lean body compartment and to the high variability among studies, in terms of inclusion criteria, measurement mode of body compositions, androgenic compounds selected, dosages, scheme and route of administration’.

8) I think that the authors should even more stress the fact that the main limitation is the limited number of studies available for body composition (only 5-6 for LBM and FFM). They should also acknowledge more clearly in the limitation that there are only short-term studies (maximum duration 6 months) and that some of the differences could not be statistical significant because of this fact.

ANSWER: Thank you for this comment. We reinforced these concepts in the discussion section as follows: ‘In addition, while only few trials were designed to assess the lean mass compartment change in HIV setting limiting data availability, the androgen therapeutic schemes were offered for a maximum of six months, debarring long-term considerations’.

Reviewer 2 Report

In this study Santi et al have conducted a systemactic review of the available scientific literature on hypogonadism in HIV infected men. In parallel they have investigated the effect of androgen administration on body composition in HIV patients.  

            The authors examined publications from two scientific literature databases, covering several decades of research on the study subject. Overall, dozens of thousands of papers were initially screened, followed by a thorough analysis of 30-40 clinical descriptive studies and therapeutic intervention trials, respectively.

In summary, the review reports a consistently high prevalence of hypogonadism and significantly overall lower levels of free testosterone in the serum of HIV-infected men. Treatment of male HIV patients with testosterone was found to have beneficial effects, including increased lean body mass and fat free mass.     

            The review is well written and convincingly presented. The covered topic is of significant scientific and clinical relevance, and will certainly have a considerable readership. The literature research methods and statistical tools used for the meta-analysis are appropriate and were efficiently used. Meta-data are moreover clearly presented and convey primary data instructively.

Overall, due to the very specific nature of the covered topic, the audience of the manuscript will be limited and the presented data are not of general interest to the broader public. Nonetheless, I believe the manuscript is absolutely worth publishing and I have no doubt it will have a substantial impact in its specific scientific field.

Some minor issues should still be addressed:

General remarks: Does the study distinguish between untreated and HAART-suppressed HIV infected individuals. If not, could the authors comment on why they think this factor should not be considered, given the impact of HAART on the overall health, but potentially also the androgen homeostasis of this patient population. If all contemplated studies are focusing on suppressed patient populations, this should be stated in the manuscript.

The figure resolution is pretty low.

Line 22: The word order is wrong in this sentence and therefore it is hard to comprehend.

Line 27 and throughout the manuscript: In the context of HIV is T deficiency a bit confusing. When reading swiftly the reader may mix it up with T cell deficiency. The authors may want to consider a different abbreviation for testosterone.

Line 102: Did the authors make sure to remove duplicates from all their analyses? What measures were taken to do that? It may be interesting to the reader why these two literature databases were chosen.

Line 181: Based on what criteria were these 69 papers selected? How were the 15562 papers excluded?

Line 198: In the table there seems to be a letter missing in the word “homosexual”. Moreover, some of the investigated publication (i.e. Arver) are not perfectly age-matched when comparing control and patient populations. Should such papers be excluded given the impact of age on Testosterone levels?

Line 204: The authors should rationalize this exclusion!

Line 208: There is a typo in the x-axis label of S2 B.

Line 211 (Figure 2): The authors may increase the fonts in this figure. Abbreviations may be introduced (LL, UL).

Line 217 and 222: The authors probably mean "significant" differences.

Line 240: Again, the authors should briefly state why 23603 papers were excluded.

Line 260: Whenever the number of trials are stated that reported a specific observation it would be nice to know out of how many.

Line 355: Did the authors state the prevalence of hypogonadism in healthy, age-matched individuals? For comparison it should be stated whenever the prevalence in HIV patients is reported.

Line 488-489: Did the authors state the prevalence of hypogonadism in healthy, age-matched individuals? For comparison it should be stated whenever the prevalence in HIV patients is reported.

Author Response

Reviewer 2

In this study Santi et al have conducted a systematic review of the available scientific literature on hypogonadism in HIV infected men. In parallel they have investigated the effect of androgen administration on body composition in HIV patients. The authors examined publications from two scientific literature databases, covering several decades of research on the study subject. Overall, dozens of thousands of papers were initially screened, followed by a thorough analysis of 30-40 clinical descriptive studies and therapeutic intervention trials, respectively.

In summary, the review reports a consistently high prevalence of hypogonadism and significantly overall lower levels of free testosterone in the serum of HIV-infected men. Treatment of male HIV patients with testosterone was found to have beneficial effects, including increased lean body mass and fat free mass. The review is well written and convincingly presented. The covered topic is of significant scientific and clinical relevance, and will certainly have a considerable readership. The literature research methods and statistical tools used for the meta-analysis are appropriate and were efficiently used. Meta-data are moreover clearly presented and convey primary data instructively.

Overall, due to the very specific nature of the covered topic, the audience of the manuscript will be limited and the presented data are not of general interest to the broader public. Nonetheless, I believe the manuscript is absolutely worth publishing and I have no doubt it will have a substantial impact in its specific scientific field.

Some minor issues should still be addressed:

1) General remarks: Does the study distinguish between untreated and HAART-suppressed HIV infected individuals. If not, could the authors comment on why they think this factor should not be considered, given the impact of HAART on the overall health, but potentially also the androgen homeostasis of this patient population. If all contemplated studies are focusing on suppressed patient populations, this should be stated in the manuscript.

ANSWER: We agree with the reviewer’s comment. In the complex scenario of the hypothalamic-pituitary-gonadic axis in HIV setting, the HAART treatment is clearly a relevant contributor. However, since almost all HIV patients considered were under HAART treatment, we are still unable to discriminate the influence induced by HAART and by HIV per se to androgens homeostasis. We specified this aspect in the results section.

2) The figure resolution is pretty low.

ANSWER: Thanks for your comment. We improved figures’ resolution.

3) Line 22: The word order is wrong in this sentence and therefore it is hard to comprehend.

ANSWER: Thank you for your comment. We rephrase the sentence as follows: ‘Moreover, in HIV setting, the efficacy of exogenous testosterone (Te) administration at improving body composition remains unclear’.

4) Line 27 and throughout the manuscript: In the context of HIV is T deficiency a bit confusing. When reading swiftly the reader may mix it up with T cell deficiency. The authors may want to consider a different abbreviation for testosterone.

ANSWER: Thank you for your comment. We change the abbreviation for testosterone from ‘T’ to ‘Te’ throughout the manuscript.

5) Line 102: Did the authors make sure to remove duplicates from all their analyses? What measures were taken to do that? It may be interesting to the reader why these two literature databases were chosen.

ANSWER: Four authors performed separately literature searches, trying to include all published articles on the considered topics. Systematic reviews are generally conducted quering these databases, in order to obtain a comprehensive view of studies available. Quality checks for selected studies were performed on the final database, including duplicates removal.

6) Line 181: Based on what criteria were these 69 papers selected? How were the 15562 papers excluded?

ANSWER: According to Cochrane Collaboration and PRISMA statement for meta-analyses conduction, we included only studies fulfilling inclusion criteria as specified in 2.1 paragraph (Study selection and inclusion criteria).

7) Line 198: In the table there seems to be a letter missing in the word “homosexual”. Moreover, some of the investigated publication (i.e. Arver) are not perfectly age-matched when comparing control and patient populations. Should such papers be excluded given the impact of age on Testosterone levels?

ANSWER: Thank you for your comments. We corrected the wrong spelling as you suggested. Moreover, we agree with the reviewer about the relevance of age when considering testosterone levels. However, we decided to not exclude Arver et al. since age-related hormonal changes develop in most men at about the age of 50 (Lunenfeld B. Review World J Urol. 2003 Nov;21(5):292-305. doi: 10.1007/s00345-003-0366-8) while enrolled men were at least ten years younger. Thus, we could reasonably assume that testosterone levels could be comparable in the two groups.

8) Line 204: The authors should rationalize this exclusion!

ANSWER: Thank you for your comment. As reported in the introduction section, several cut-offs for hypogonadal definition were proposed over the years and the same heterogeneity emerged in included studies. However, the vast majority of randomized controlled trials and meta-analyses considered testosterone cut-off ranging from 10 to 12 nmol/l (Rastrelli G, Guaraldi F, Reismann Y, Sforza A, Isidori AM, Maggi M, Corona G. Sex Med Rev. 2019 Jul;7(3):464-475. doi: 10.1016/j.sxmr.2018.11.005). Thus, we decided to include all studies which considered the definition of hypogonadism with a threshold below 12 nmol/l, i.e. the less stringent cut-off, to increase data availability.

9) Line 208: There is a typo in the x-axis label of S2 B.

ANSWER: Thank you. We corrected the figure.

10) Line 211 (Figure 2): The authors may increase the fonts in this figure. Abbreviations may be introduced (LL, UL).

ANSWER: Thank you. We introduced the lacking abbreviations.

11) Line 217 and 222: The authors probably mean "significant" differences.

ANSWER: Thank you for this comment. We added “significant” to these differences.

12) Line 240: Again, the authors should briefly state why 23603 papers were excluded.

ANSWER: We better explained the reason for exclusion as follows: ‘The second literature search identified 23636 papers, and 33 studies were included in the full-text analysis, since they fulfilled inclusion criteria (Figure 1). After the full-text evaluation, 17 studies were finally considered (Figure 1 and Table 2). Among the 12 trials excluded during the full text evaluation, four trials did not contemplate a control group, four trials did not report body composition parameters, seven trials considered together both male and female patients and one provided a previous Te administration in both groups before androgen supplementation (Table 2) (Figure 1).’

13) Line 260: Whenever the number of trials are stated that reported a specific observation it would be nice to know out of how many.

ANSWER: Thank you for this comment. The number of studies analysed in the meta-analytic approach, i.e. in each specific observation, was reported in the associated figure. We did not report these numbers within the manuscript to make the text smoother when reading.

14-15) Line 355, 488-489: Did the authors state the prevalence of hypogonadism in healthy, age-matched individuals? For comparison it should be stated whenever the prevalence in HIV patients is reported.

ANSWER: Our review is focused on the analysis of hypogonadism in HIV-infected men. We did not report the prevalence of hypogonadism in healthy age-matched individuals since only five studies out of 12 calculated this percentage, which ranged from 0 to 46%. We think that such limited and variable data did not provide a reliable result for an effective comparison.

Reviewer 3 Report

Review article by Santi et al. is very well written. Overall, it provides an interesting summary of the issue. The statistics seem to be well processed and the links in the text are supplemented in detail by pictures in the appendix.

1) In the introduction, I miss the definition of androgens as chemical entities. From my point of view, it is necessary to define androgens as steroids (which does not appear in the article) with a suitable citation (e.g. 10.3390/molecules26041032).

2) In the abstract, the materials and methods section seems redundant to me.

3) P1L39: Androgen replacement treatment or therapy?

4) Table 1 and Table 2: Maybe it would be better to make the landscape page layout. It doesn't look very nice like this. For better readability, I would add the citation number to which the data relate. (Selehian - different font size).

5) Please complete the existence of Supplementary in the main article and describe the supplementary figures as in the guidelines for authors.

6) Literature should be homogeneously formatted according to the guidelines for authors.

Overall, I rate this paper as very well written and beneficial to the community of endocrinologists and physicians.

Author Response

Reviewer 3

Review article by Santi et al. is very well written. Overall, it provides an interesting summary of the issue. The statistics seem to be well processed and the links in the text are supplemented in detail by pictures in the appendix.

1) In the introduction, I miss the definition of androgens as chemical entities. From my point of view, it is necessary to define androgens as steroids (which does not appear in the article) with a suitable citation (e.g. 10.3390/molecules26041032).

ANSWER: We better specified in the introduction the definition of androgens as chemical entities, citing proper bibliography, as suggested.

2) In the abstract, the materials and methods section seems redundant to me.

ANSWER: The materials and methods section in the abstract is required by the journal rules. Thus, we have to maintain it.

3) P1L39: Androgen replacement treatment or therapy?

ANSWER: Thank you fo your comment. We referred to androgen replacement treatment throughout the manuscript.

4) Table 1 and Table 2: Maybe it would be better to make the landscape page layout. It doesn't look very nice like this. For better readability, I would add the citation number to which the data relate. (Selehian - different font size).

ANSWER: Thank you for your comment. The tables layout was provided by the editor, and not by the authors directly.

5) Please complete the existence of Supplementary in the main article and describe the supplementary figures as in the guidelines for authors.

ANSWER: Supplementary tables and figures are reported within the text, when required.

6) Literature should be homogeneously formatted according to the guidelines for authors.

ANSWER: Thank you for your comment. We formatted the manuscript accordingly.

Overall, I rate this paper as very well written and beneficial to the community of endocrinologists and physicians.

ANSWER: Thank you for your appreciation.